# Bracket Bond Failures: Incidence and Association with Different Risk Factors—A Retrospective Study

**DOI:** 10.3390/ijerph20054452

**Published:** 2023-03-02

**Authors:** Reda Jakavičė, Kristina Kubiliūtė, Dalia Smailienė

**Affiliations:** Department of Orthodontics, Faculty of Odontology, Lithuanian University of Health Sciences, 44307 Kaunas, Lithuania

**Keywords:** bonding failure, bracket failure, risk factors

## Abstract

Background: Bracket bonding failure is one of the relevant problems in fixed orthodontics therapy, which affects the total treatment and quality of treatment results. The purpose of this retrospective study was to evaluate the frequency of bracket bond failure and find out risk factors. Methods: A total of 101 patients with an age range of 11–56 years were included in this retrospective study and treated for a mean period of 30.2 months. Inclusion criteria were: males and females with permanent dentition and completed orthodontic treatment in both fully bonded dental arches. Risk factors were calculated using binary logistic regression analysis. Results: The overall bracket failure rate was 14.65%. The bracket failure rate was significantly higher in the younger patients’ group (*p* = 0.003). In most cases, patients experienced bracket failures in the first month of the treatment. Most of the bracket bond failures occurred on the left lower first molar (29.1%) and were twice as common in the lower dental arch (66.98%). Patients with increased overbite had an increased likelihood of bracket loss (*p* = 0.042). Class II malocclusion increased the relative risk of bracket failure, while Class III decreased the rate of bracket failure, but the difference was not statistically significant (*p* = 0.093). Conclusions: The bracket bond failure rate was higher in younger patients than in older patients. Brackets placed on mandibular molars and premolars had the highest failure rate. Class II was associated with an increased bracket failure rate. Increased overbite statistically significantly increases bracket failure rate.

## 1. Background

Malocclusion is known as one of the most common disorders of oral and maxillofacial areas [1]. Misalignment of teeth is prevalent in between 39% and 93% of children and adolescents [2]. Based on different research data, the most prevalent orthodontic anomalies are Class I (29.5% to 70.22%), followed by Class II, while Class III is the least common [3,4]. Misalignment of teeth can predispose to periodontal disorders [5] and temporomandibular disorders [6] and also have a negative influence on oral health-related quality of life [7]. Nowadays, orthodontic therapy improves the quality of life by effectively restoring malocclusions, optimal chewing function, and dentofacial aesthetics. These aims of orthodontic treatment can be perfectly achieved by using bracket systems. The dental brackets bonding procedure to enamel is very important for achieving a satisfactory outcome of orthodontic treatment because dental movement depends on the quality and strength of bonding. However, bracket bonding failure is one of the inevitable problems in fixed orthodontics therapy, which according to the literature, occurs in 0.6–17.1% of all brackets [4,8,9,10,11,12]. 

Bracket debonding is one of the main factors for repeated emergencies. Besides, this problem potentially increases total treatment time, chairside time and financial charge of the treatment, as well as compromises treatment results and may cause damage to enamel [4,12,13]. According to the results of the Stasinopoulos et al. study, every bracket rebonded due to failure can increase the treatment duration by about 0.6 months [9].

Failure of brackets is a multifactorial problem. The main reasons for brackets failure are associated with operator, patient and material factors. Debonding of braces can occur due to many patients-related factors such as existing malocclusion, oral hygiene, bad habits, patient compliance, age, masticatory force, some conservative restorative dentistry therapies or bleaching procedures performed prior to bonding, and operator-related factors like bonding protocol, type of orthodontic adhesive, curing lamps, brackets, and wires properties [4,8,12,13,14,15,16,17,18,19].

Numerous studies have been done to sort out different factors associated with bracket failure during orthodontic treatment. Most of these studies evaluated the influence of age, gender, tooth type, type of brackets and bonding material on bond failure [4,8,9,10,12,13,20,21,22,23,24]. In some studies, the impact of malocclusion in terms of Angle class [4,8] and overbite [8,9] on bond failures was evaluated. Stasinopoulos et al. [9] investigated the influence of crossbite and anterior crowding on the frequency of bracket failure. However, the connection between the overjet, detailly evaluated anterior teeth crowding and bond failure has not been clarified. Most of these studies were limited by a small sample size [10,13,22,23,24] or short patient observation time (6–12 months) [8,10,13,21,22,23,24].

The purpose of this retrospective study was to evaluate the frequency of bracket bond failure and risk factors, such as bad oral hygiene, increased overbite and overjet, and tooth crowding in patients who underwent orthodontic treatment.

## 2. Materials and Methods

### 2.1. Data Collection

The ethical approval for the retrospective study was obtained from the Bioethics Center (protocol number BEC-OF-139). This study was performed in the Department of Orthodontics, Faculty of Odontology.

The sample size was calculated by the calculator for prevalence studies using a formula without finite population correction [25]. Considering a confidence level of 95%, the precision was 5%, and the expected prevalence was 3.43% based on Kafle et al. results [4], the minimum sample size required was 51 subjects.

The orthodontic records of 101 completed treatment cases were selected from the archive, in which patients were treated by two orthodontics (authors DS and KK) with more than ten years of work experience. Inclusion criteria were: both males and females; permanent dentition; patient treated only with fixed appliances in both arches; fully bonded dental arches; orthodontic treatment completed; complete documentation (clear medical history, pre-treatment patients dental cast and photos available); patients with all types of malocclusion; both extraction and non-extraction cases. Exclusion criteria: patients with enamel defects, hypoplasia, big dental restorations and orthognathic surgery cases.

Operators bonded all the brackets using the same following protocol: (1) cleaning using a rubber cup with pumice without fluoride, rinsing, isolating with cheek retractors and a low-volume suction evacuator, drying, (2) etching with 38% phosphoric acid for 15 s, thoroughly rinsing with water for 10 s and drying with oil-free air until the enamel has become frosty white (etch and rinse approach), (3) application of light-cure High-Q-Bond Bracket primer to the etched enamel according to the manufacturer’s instructions, a gentle burst of air, (4) bonding with light-cure High-Q-Bond Bracket adhesive (B.J.M. Laboratories Ltd., Yehuda, Israel). The brackets were positioned along the long axis of the teeth with the help of a bracket positioning gauge. Pressure was applied to squeeze out excess adhesive, which was removed from the margins of the bracket base before polymerisation, (5) curing with a light-curing system with standard specifications (20 s).

Collected data: age at the beginning of treatment and sex, dates of start and completion of treatment and bracket loss of each patient. The brackets rebonded by the orthodontist in order to improve the position were not included in the calculation. Subsequently, Angle malocclusion, overjet, overbite and dental crowding were determined by evaluating patients’ intraoral photographs and dental models from OnyxCeph^3TM^ software. Crowding was evaluated on lower teeth and considered as mild with a discrepancy of less than 4 mm; a discrepancy between 5 mm to 8 mm was determined as moderate crowding, and severe crowding was determined when there was a discrepancy of more than 8 mm. The normal overjet and overbite range was considered to be 2–3 mm. The overjet and overbite of more than 4 mm were regarded as increased, while less than 1 mm was considered as decreased. Information about patients’ dental hygiene was received from medical records and intraoral photographs during orthodontic treatment.

### 2.2. Statistical Analysis

Statistical analysis was performed using SPSS Statistics software (Version 29, IBM, New York, NY, USA). Descriptive statistics were reported. The interdependence of qualitative characteristics was evaluated using the chi-squared (χ^2^) criterion. The Kolmogorov–Smirnov test was used to determine the normality of the parameter distribution. If the variable did not meet the distribution normality condition, a significance level between three independent groups was verified by the nonparametric Kruskal–Wallis test. The probability of the event given a certain risk factor was calculated using binary logistic regression analysis, including an odds ratio (OR) and its confidence interval (95% CI). Spearman’s correlation test was used for the analysis of significant relations. The level of statistical significance for all tests was set at *α* = 0.05.

## 3. Results

### 3.1. Study Sample

A total of 101 patients were included in the final analysis according to the inclusion criteria. Of them, 73 (72.3%) were females, and 28 (27.7%) were males, with ages ranging from 11 to 56 years old. The mean age of the included patients was 20.0 years, and most of them were under the age of 20 years (n = 68). The characteristics of the included patients are presented in Table 1.

Based on data, 84 patients maintained good oral health during orthodontic treatment, while 17 patients had complicated oral health conditions, including bad oral hygiene and gingivitis recorded in medical records. The characteristics of the included patients‘ malocclusion are presented in Table 2.

### 3.2. Failure Rate and Patients-Related Factors

A total of 2791 teeth were bonded with brackets and tubes. The overall bracket failure rate was 14.65% (n = 409), and 78.22% of the patients had an incidence of bracket failure (n = 79). On average, four brackets were debonded per patient (range 0–37).

Most of the bracket failure occurred on the left lower first molar (12.47%), followed by the right lower first molar (11.74%), while the lower left lateral incisor reported had the lowest frequency (0.49%). Bond failure was twice as common in the lower dental arch (66.98%) in comparison to the upper dental arch (33.02%) (*p* = 0.003) (Figure 1). In Class I and Class II cases, the brackets mostly debonded from the lower first molars, while in Class III cases, they were from the lower first and second premolars.

The chi-squared test showed that the bracket failure rates between the genders and oral health conditions were shown not to be significant (*p* = 0.095, *p* = 0.065) (Table 3). Binary logistic regression analysis revealed a significant link between bracket failure and different age groups. Patients younger than 20 years old had an increased risk of bracket failure (OR 4.26; CI:1.58, 11.46; *p* = 0.003). The ROC curve was used to predict the value of the threshold age (Figure 2). 

### 3.3. Malocclusion-Related Factors

A Kruskal–Wallis test showed that there was not a statistically significant difference in bracket failures between the different classes of Angle (*p* = 0.396), with a mean rank bracket failure of 48.02 for Class I, 56.16 for Class II and 48.00 for Class III. Data showed that the average number of bracket failures was highest in patients with Class II. (Table 4).

Binary logistic regression analysis was then performed to control for possible confounding by any of the aforementioned variables that reached the predetermined level of significance of *p* < 0.05 (Table 5). The variables included in the model based on this criterion were Angle class, overbite, overjet, and crowding of teeth. The *p*-values for the deviance and Pearson Chi-squares are all larger than 0.05, and although there is a little over-dispersion, this model seems to have an acceptable fit with the data.

Based on the data, Class II malocclusion increased the relative risk of bracket failure but was not statistically significant (*p* = 0.093), while Class III decreased the rate of bracket failure. In the case of an overbite, a clear trend emerged in which patients with increased overbite had an increased likelihood of bracket loss (*p* = 0.042). Also, overjet was not significant for any category.

### 3.4. Treatment-Related Factors 

In most cases, patients experienced bracket failures in the first month of the treatment, followed by the third, fourth and sixth months (Figure 3).

Likewise, the bracket failure rates between the two different operators and type of brackets were shown not to be significant (*p* = 0.778, *p* = 0.392).

## 4. Discussion

In this retrospective study, the overall bracket failure rate was 14.65%, which is higher compared with some studies [4,8,13,20], where the bracket failure rate ranged from 3.43% to 6.4%. On the contrary, Aikins et al. study reported a higher bracket failure rate (17.1%) [12]. Similar results of bracket failure rate during the whole course of orthodontic treatment (14.1%) were reported by Stasinopoulos et al. study [9]. It seems to be that studies with longer observation periods show higher bracket bond failure rates.

In this study, younger patients demonstrated a statistically significant higher bracket failure rate than older patients, which is in agreement with Jung et al. study [20]. In contrast, Khan et al. and Sakrani et al. studies showed no significant difference in bond failure with patients’ age [8,21]. Of the bracket failures, 58.4% occurred in younger patients (≤20 years). Various reasons could lead to a higher failure incidence in younger patients, such as careless behaviour and less self-motivation, whereas adults are more compliant with given orthodontic instructions. The reason for poor patient cooperation could also be related to the present payment system, in which care is provided at no direct cost to patients under 18 years. Also, thick, gingival biotype and partial eruption of lateral teeth and habits may also predispose higher failure incidence in younger patients.

Debonding of brackets in this study was more commonly found in the mandibular dental arch (*p* < 0.01). Such results correspond to the previous studies carried out by other researchers [8,12,21,22,23]. These higher failure rates were attributed to occlusal interferences because brackets on lower teeth can be affected by upper teeth cups during mastication, especially in case of increased overbite. Nevertheless, Naqvi et al.’s study provided the opposite results [24]. Present findings are consistent with previous studies, which showed that bond failure is more common in posterior teeth [8,9,20]. It is probable that posterior teeth are more difficult to isolate from moisture during the bonding of brackets. Moreover, posterior teeth are affected by heavier occlusal forces than anterior teeth during mastication [22]. It is worth mentioning that the bonding strength could be affected by greater aprismatic enamel and poorer etch quality for the posterior teeth than the anterior teeth [26]. Besides, first molars have big restorations and attrition more often, which worsen the anatomy of the tooth and lead to inadequate adaptation of the bracket to the tooth surface.

Based on these results, the highest number of debonding brackets were observed in patients with Class II, but the differences were not statistically significant. The statement that Class II malocclusion increases the likelihood of bracket failure is confirmed by many authors [4,8,14,22]. Class II malocclusions are often associated with increased overjet and overbite. Therefore, brackets are in contact with teeth during mastication. Except for crowding, increased overbite was the main occlusal factor statistically significantly associated with increased bond failure. 38.61% of the bracket failures occurred in cases of deep bite, in comparison to 35.64% of normal and 25.74% of decreased overbite. Khan et al. also stated that the failure rate in patients with normal overbite was 41.1%, the failure rate in cases with decreased overbite was 15%, while in deep bite cases, the failure rate was 43.9%, with a statistically significant difference [8]. The importance of overbite for brackets debonding rate is also stressed by Stasinopoulos et al. study [9]. Furthermore, patients with deep bites often have hyperactivity of masseters and anterior temporalis muscles, especially in growing children, which could lead to the heavier force of mastication [27]. For this reason, bite-raising must be performed in order to eliminate occlusal interferences and prevent the brackets from being debonded from the beginning of treatment.

While assessing the effect of oral hygiene status of patients treated with fixed orthodontic appliances, significant differences were not found in the bond failure rate. However, the literature suggests that patients with poor hygiene are more prone to bracket failure. Higher bracket failure could possibly be explained by the increased plaque accumulation, which prevents the washing out of acid from the tooth’s surface, consequently reducing the pH of the oral cavity. Decreased pH increases the risk of demineralization of dental hard tissues, which could lead to the weakening of the bracket’s adhesion force [28].

According to data, 55.98% of brackets debonding was observed in the first year of the orthodontic treatment, while 32% of brackets failed in the second year and only 12.02% in the third year. The statement that bracket failure is more common in the first twelve months is confirmed by other studies [8,12]. The decreasing tendency of bracket failures in later stages of treatment could be due to several reasons, such as the adaptation of patients, increasing patient age and motivation, as well as bite correction, which normally occurs during the first year of treatment. It can also be assumed that braces that had not been bonded properly debonded during the first months of treatment.

In many studies, the observation period of bracket debonding was from six months to 24 months [4,8,13,20,22,23], whereas, in the present study, the failure rates were evaluated throughout the whole treatment period. Sample size calculation was performed, and the study sample was larger than in many other studies. Lastly, many variables, including overjet, deep bite, crowding and malocclusion classification were analysed, which helps better predict the reasons for the debonding of the brackets. However, several limitations were present. Firstly, the retrospective design of the study. All included patients were treated by two clinicians whose approach to working could be different, and it might have influenced the study’s results. However, the bracket failure rates between the two different operators were shown not to be significant. Moreover, bracket failure may have been influenced by different treatment mechanics. Likewise, this study included patients with different anomalies, such as impacted teeth or hypodontia, extraction and non-extraction cases, and it could also have an influence on results due to orthodontic treatment features. 

This study aimed to evaluate patients-related factors of bracket bond failure. However, there are other operator and material-related factors like bonding protocol, type of orthodontic adhesive, brackets, and wires properties [4,8,12,13,14,15,16,17,18,19] that can have an impact on bond strength. Ayyed et al. and Thiyagarajah et al. compared bracket failure rates between direct and indirect bonding techniques and did not find clinically significant differences in the bond failure prevalence [15,16]. The quality of the brace bonding procedure can also depend on bonding material. According to the literature the required bond strength of adhesive systems ranges between 5.9 to 7.8 MPa, and it is sufficient to withstand masticatory forces [29]. The conventional bonding protocol was used in this study; however, it is very important to evaluate and compare the orthodontic bracket failure patterns using different bonding protocols. Results from the studies indicate that conventional light-cured bonding resin (Transbond XT) has significantly higher strength than resin-modified glass ionomer (Fuji Ortho LC) [17,18] or ACP-amorphous calcium phosphate bonding system [22]. In vitro study has been carried out to evaluate bonding strength while comparing flowable with conventional composites. The study concluded that the conventional one exhibited higher bond strength [30]. In clinical practice, the bond failure rates between the conventional two-step “etch and primer” method and the self-etching primer method were shown not to be significant [31,32]. However, Sabatini et al. show that self-etching primers produce lower bond strength (18.3 ± 6.7 MPa) than the traditional combination of phosphoric acid etching and bonding resin (37.7 ± 3.2 MPa) [33]. There is considerable debate over the appropriateness of primer use. Studies evaluating bracket failure bonded with and without primer showed no significant difference during orthodontic treatment [34,35].

In addition, the bond failure modes are classified into the adhesive-enamel (debonding occurs between the adhesive and the enamel surface), adhesive-bracket (debonding between the adhesive and the bracket) and cohesive failure (debonding within the adhesive itself). The type of bracket failure might be influenced by various factors such as the mechanical properties of the adhesive resin, morphology of the bracket base and the bond strength values achieved with the use of adhesive systems [36]. Bonding materials with a low bond strength to enamel tend to debond at the adhesive/enamel interface, whereas materials with high enamel—resin bond strength tend to show cohesive failures or adhesive/ bracket debonding. Cases of adhesive-bracket fracture might be caused by little penetration of the primer due to the reduced enamel demineralization [37]. Usually, the use of conventional bonding techniques shows cohesive bond failure [37,38].

Also, different types of brackets may have different resistance to debonding. Patients included in this study were treated with two types of brackets: MBT 0.022 Forestadent and Damon Q2. Bracket failure rates between the two types of brackets were shown not to be significant. The results of studies assessing the relationship between different types of brackets and bond failure rates are controversial. Oginski et al. reported that metal brackets demonstrated significantly higher failure rates than ceramic brackets [14]. Ceramic brackets show significantly higher debonding strength than metal brackets [19]. However, Stasinopoulos et al. reported that ceramic conventionally ligated brackets were 60% more prone to failure when compared with metal brackets [9]. Self-ligating Damon brackets do not demonstrate statistically significant different failure rates relative to conventional brackets [39].

In addition, studies show that the type of archwire used may have an influence on bond failure. According to a study done by Khan H et al., the majority of brackets (41.1%) failed on the 0.016 NiTi wires, while 30% of brackets failed on regular NiTi wires and only 28.9% on stainless steel wires [8].

From the results of the present study, it can be said that the most important actions in order to prevent debonding are bite-raising to eliminate occlusal interferences and careful verbal and written instructions to each patient regarding appliance care. The motivation of the patient is one of the reasons for better cooperation and a lower rate of bracket failure.

## 5. Conclusions

The brackets bond failure rate was higher in younger patients (age < 20 years) than in older patients. Brackets placed on mandibular molars and premolars had the highest failure rate. This study did not find any significant difference in debonding rate depending on malocclusion and overjet, although Class II was associated with an increased bracket failure rate. Increased overbite statistically significantly increases the bracket failure rate.

## Figures and Tables

**Figure 1 ijerph-20-04452-f001:**
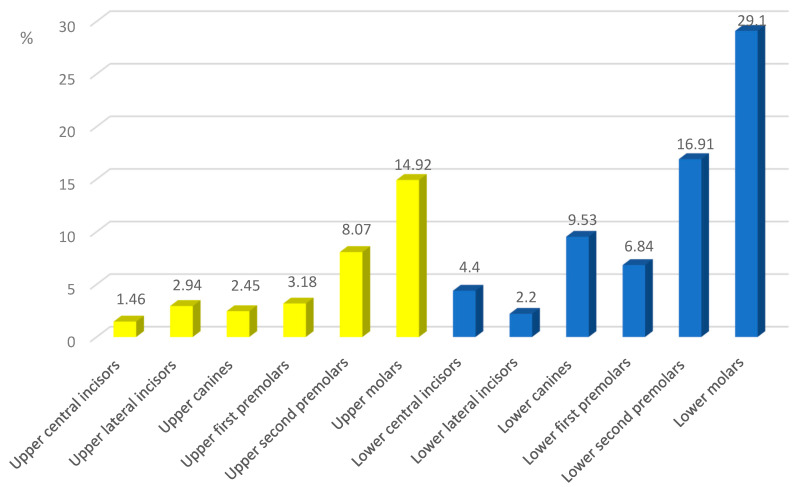
Distribution of brackets failures in different groups of teeth.

**Figure 2 ijerph-20-04452-f002:**
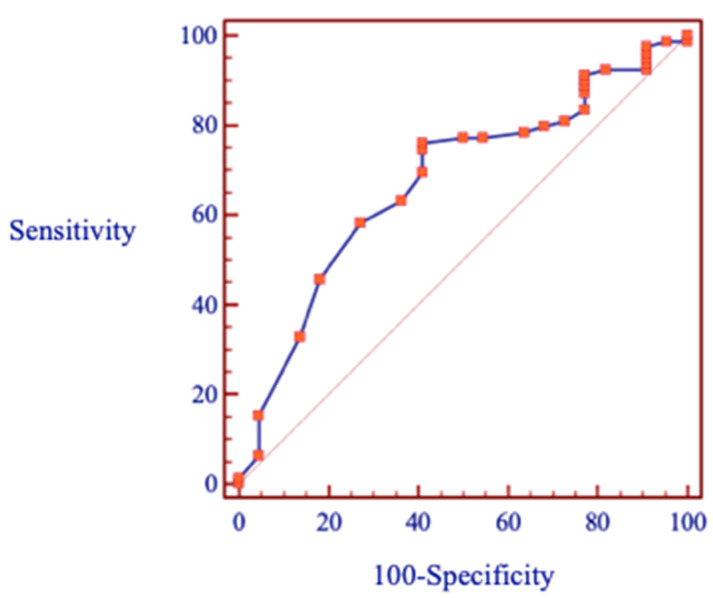
The receiver operating characteristic (ROC) curve to evaluate trade-off between sensitivity and specificity. The area under the curve—67.5%, sensitivity—75.9%, and specificity—59.1%.

**Figure 3 ijerph-20-04452-f003:**
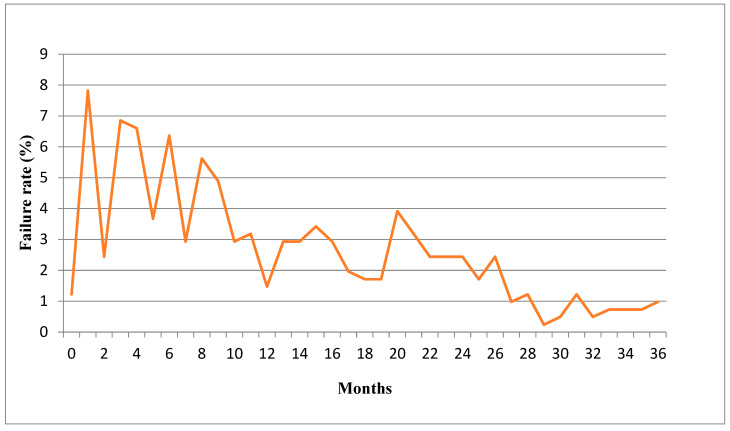
Bracket failure rates in different stages of orthodontic treatment.

**Table 1 ijerph-20-04452-t001:** Descriptive characteristics of the patients’ sample included in the present study.

Variable	
Gender, n (%)	
Male	28 (27.7)
Female	73 (72.3)
Age at start of orthodontic treatment (years)	
Mean	20.0
≤20 years, n (%)	68 (67.3)
>20 years, n (%)	33 (32.7)
Type of bracket system, n (%)	
MBT 0.022 Forestadent	54 (53.5)
Damon Q2	47 (46.5)
Duration of orthodontic treatment (months)	
Mean	30.2 (range 12–76)

**Table 2 ijerph-20-04452-t002:** Characteristics of included patients‘ malocclusion at baseline.

Variable	
Angle class of molars at the start of orthodontic treatment, n (%)	
Class I	49 (48.5)
Class II	37 (36.6)
Class III	15 (14.9)
Overjet, n (%)	
Decreased overjet	28 (27.7)
Normal overjet	32 (31.7)
Increased overjet	41 (40.6)
Overbite, n (%)	
Decreased overbite	26 (25.7)
Normal overbite	36 (35.6)
Increased overbite	39 (38.6)
Crowding mandibular arch, n (%)	
Mild (0–4 mm)	45 (44.6)
Moderate (5–8 mm)	22 (21.8)
Severe (>8 mm)	17 (16.8)
Space excess	17 (16.8)

**Table 3 ijerph-20-04452-t003:** Comparison of bond failures among gender, age and oral health conditions.

		Failure	Non-Failure	*p*
Gender, n (%)	Male	25 (89.3)	3 (10.7)	0.095
	Female	54 (74.0)	19 (26.0)	
Age, n (%)	≤20	59 (86.8)	9 (13.2)	**0.003** **
	>20	13 (39.4)	20 (60.6)	
Oral health, n (%)	Complicated	14 (82.4)	3 (17.6)	0.065
	Non-complicated	65 (77.4)	19 (22.6)	

*p* values from Chi-square test; ** *p* < 0.05.

**Table 4 ijerph-20-04452-t004:** Summary statistics of the non-normally distributed outcome number of failed brackets for different variables.

Factor	Category	n	Mean Rank	df	*p* ^a^	χ^2^
Angle class	Class I	49	48.02	2	0.396	1.851
	Class II	37	56.16			
	Class III	15	48.00			
Overbite	Increased	39	52.13	2	0.381	1.930
	Normal	36	54.53			
	Decreased	26	44.42			
Overjet	Increased	41	57.27	2	0.119	4.250
	Normal	32	43.17			
	Decreased	28	50.77			
Crowding	Mild	45	43.58	2	0.903	0.204
	Moderate	22	41.59			
	Severe	17	40.82			

^a^*p* values from Kruskal–Wallis tests.

**Table 5 ijerph-20-04452-t005:** Binary logistic regression analysis to detect risk factors of bracket failure.

Factor	Category	N	Coefficient	Odds Ratio (95% CI)	*p*
Angle class	Class I	49	–	–	0.200
	Class II	37	1.162	0.313 (0.081–1.213)	0.093
	Class III	15	−0.811	0.444 (0.075–2.619)	0.370
Overbite	Increased	39	1.434	4.196 (1.055–16.692)	**0.042** *
	Average	36	–	–	0.081
	Decreased	26	−0.048	0.953 (0.168–5.407)	0.957
Overjet	Increased	41	–	–	0.078
	Average	32	1.404	4.073 (0.811–20.453)	0.088
	Decreased	28	−0.732	0.481 (0.099–2.335)	0.364
Crowding	Mild	45	1.187	3.277 (0.438–24.508)	0.248
	Moderate	22	2.538	12.656 (1.388–115.442)	**0.024** *
	Severe	17	1.539	4.660 (0.510–42.611)	0.173
	Space excess	17	–	–	0.100

* *p* < 0.05.

## Data Availability

The datasets used and/or analyzed during the current study are available from the corresponding author on reasonable request.

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
