# Peer review of "Bracket Bond Failures: Incidence and Association with Different Risk Factors—A Retrospective Study"

_ijerph, 2023, doi:10.3390/ijerph20054452_

Round 1

Reviewer 1 Report

This is a retrospective look of debonding failures of a single orthodontic clinic, therefore, conclusions should be not extrapolated in general. However, it is an interesting reading, and source of comparison to previously published literature. Yet, there are some important aspects that should improve readability and understanding.
There is no mention of the method or protocols for bonding implied. Was it GI, composite, or else, how teeth were isolated, what kind of adhesive was used, was it self-etch or etch and rinse approach, how many steps utilized, what was the pH and the hydrophilicity of the layer before composite (if so).
Additionally it would be essential to know where was the bond failure occuring. Adhesive failure in enamel, in bracket side or cohesive on the composite. Those are factors signifying lots of conclusions may occur for bonding.
Also, what were the procedures before bonding of the enamel, were the teeth cleaned with pumice containing fluoride, or no fluoride, was the bonding performed without any prior cleaning of the surface, sandblasting implied, or some alternative celaning method?
In my opinion, since all patients were treated in the same clinic, exact description of bonding protocols used is doable and advisable. Will help better understand results and draw conclusions.
Further comments on the manuscript is some language mistakes, mostly due to carelessness during edits. Examples are the mistaken use of authors name in line 71, and the use of "or" in line 132. Further such minor mistakes are noticeable as well, a futher professional edit is advised.

Reviewer 2 Report

Congratulations to the authors for this work. However, there are some questions that need to be answered.

1. In the introduction you should talk more about the influence of the type of brackets and adhesion forces. 

2. The material and method section does not describe the bonding protocol used or the type of cement. 

3.  He has used two types of brackets only, Damon and conventional MBT. He does not think it is appropriate to compare more types of brackets.

4. In the discussion section there should be more debate, with reference to bracket bonding forces and different protocols. Facial pattern and chewing forces are factors to be taken into account in the present study.

Best Regards

Round 2

Reviewer 2 Report

Dear Authors,

The paper has been improved. The questions had been responde, so in my opinion the article is suitable for publication.